# Implementing a Community-Led Arsenic Mitigation Intervention for Private Well Users in American Indian Communities: A Qualitative Evaluation of the Strong Heart Water Study Program

**DOI:** 10.3390/ijerph20032681

**Published:** 2023-02-02

**Authors:** Darcy M. Anderson, Annabelle Black Bear, Tracy Zacher, Kelly Endres, Ronald Saxton, Francine Richards, Lisa Bear Robe, David Harvey, Lyle G. Best, Reno Red Cloud, Elizabeth D. Thomas, Joel Gittelsohn, Marcia O’Leary, Ana Navas-Acien, Christine Marie George

**Affiliations:** 1Department of International Health, Johns Hopkins Bloomberg School of Public Health, Baltimore, MD 21205, USA; 2Missouri Breaks Industries Research, Cheyenne River Sioux Tribe, Eagle Butte, SD 57625, USA; 3Indian Health Service, Rockville, MD 20857, USA; 4Environmental Resource Department, Oglala Sioux Tribe, Pine Ridge, SD 57770, USA; 5Department of Environmental Health Science, Mailman School of Public Health, Columbia University, New York, NY 10032, USA

**Keywords:** arsenic, drinking water, process evaluation, consolidated framework for implementation research (CFIR), RANAS model

## Abstract

Arsenic is a naturally occurring toxicant in groundwater, which increases cancer and cardiovascular disease risk. American Indian populations are disproportionately exposed to arsenic in drinking water. The Strong Heart Water Study (SHWS), through a community-centered approach for intervention development and implementation, delivered an arsenic mitigation program for private well users in American Indian communities. The SHWS program comprised community-led water arsenic testing, point-of-use arsenic filter installation, and a mobile health program to promote sustained filter use and maintenance (i.e., changing the filter cartridge). Half of enrolled households received additional in-person behavior change communication and videos. Our objectives for this study were to assess successes, barriers, and facilitators in the implementation, use, and maintenance of the program among implementers and recipients. We conducted 45 semi-structured interviews with implementers and SHWS program recipients. We analyzed barriers and facilitators using the Consolidated Framework for Implementation Research and the Risks, Attitudes, Norms, Abilities, and Self-regulation model. At the implementer level, facilitators included building rapport and trust between implementers and participating households. Barriers included the remoteness of households, coordinating with community plumbers for arsenic filter installation, and difficulty securing a local supplier for replacement filter cartridges. At the recipient level, facilitators included knowledge of the arsenic health risks, perceived effectiveness of the filter, and visual cues to promote habit formation. Barriers included attitudes towards water taste and temperature and inability to procure or install replacement filter cartridges. This study offers insights into the successes and challenges of implementing an arsenic mitigation program tailored to American Indian households, which can inform future programs in partnership with these and potentially similar affected communities. Our study suggests that building credibility and trust between implementers and participants is important for the success of arsenic mitigation programs.

## 1. Introduction

Arsenic is a naturally occurring toxicant that is prevalent in groundwater throughout the United States. Arsenic is associated with a variety of adverse health effects, including impaired neurological function, cancer, diabetes, and cardiovascular disease [1,2,3,4]. The United States Environmental Protection Agency (EPA) set a maximum contaminant level (MCL) of 10 μg/L for arsenic in public drinking water in 2001, lowering it from the previous MCL of 50 μg/L. However, arsenic in private wells is unregulated, and testing and mitigation are the responsibility of the homeowner [5]. Since the EPA adopted its current MCL, exposure to arsenic among public water system users has decreased substantially, while exposure among private well users has not [6].

American Indian communities are disproportionately affected by arsenic in private wells [7,8,9]. Exposure occurs through drinking arsenic contaminated water or preparing foods and beverages with it (e.g., rice, pasta, juice concentrate, and powdered drink mixes, which are commonly reported food items [10]). Groundwater concentrations of arsenic in the Great Plains region are naturally elevated, and mining has further increased concentrations in some areas [11,12,13]. Disproportionately high arsenic exposure occurs in this region in part because communities are rural and water utility infrastructure is limited, so many households rely on unregulated private wells for potable water [9,11,14]. Programming for arsenic awareness and mitigation is also limited [15].

The Strong Heart Water Study (SHWS) is the first arsenic mitigation program designed to reduce arsenic exposure among private well users in American Indian communities [15]. The SHWS program was developed using a community-centered approach, and community members led program implementation. This program was delivered in the Great Plains region and included a community water arsenic testing program for private wells, installing point-of-use (POU) arsenic filters, and providing behavior change communication to promote use and maintenance of the POU arsenic filters through a mobile health (mHealth) program and in-person visits by a health promoter.

The SHWS program was evaluated through a randomized controlled trial (RCT) conducted between 2018 and 2021. Eligible households were randomized into one of two arms: “mHealth/filter” or “intensive”. The mHealth/filter arm received water arsenic testing through the community water arsenic testing program, POU arsenic filter installation by a community plumber, and the mHealth behavior change communication delivered by a community health promoter. The intensive arm received these same components, plus additional in-person behavior change communication and videos by a community health promoter [15]. Results from the RCT of the SHWS program found that both arms were highly effective in reducing arsenic exposure, evidenced by a 48% reduction in urinary arsenic (a biomarker of arsenic exposure) from baseline to the 2-year follow-up in both the mHealth/filter and intensive arms [16].

In this study, we report a qualitative evaluation of the implementation of the SHWS program. Qualitative evaluations of implementation, such as the one we report here, are important for understanding how and why trial results are achieved. They can also improve programmatic relevance of RCTs by identifying the underlying factors driving implementation and intervention success or failure, and identifying opportunities to improve or replicate trial results in other settings [17,18,19,20].

Our specific objectives were to (Objective 1) assess implementers’ and recipients’ perceptions of successes and challenges within the implementation of the SHWS arsenic mitigation program, (Objective 2) identify barriers and facilitators in implementing the arsenic mitigation program among implementers, and (Objective 3) identify barriers and facilitators in using and maintaining the POU arsenic filter among recipients.

## 2. Materials and Methods

In this study, we used qualitative methods, conducting semi-structured interviews with implementers and recipients. We also applied conceptual frameworks from implementation and behavioral science [21,22,23] to evaluate successes, barriers, and facilitators in implementation and intervention uptake. Specific methods are described below.

### 2.1. Conceptual Frameworks

We assessed perceptions of success and challenges (Objective 1) as the extent to which implementers and recipients perceived that the SHWS program had achieved the following implementation outcomes [23]: adoption, acceptability, appropriateness, cost, feasibility, fidelity, penetration, and sustainability. Definitions are provided in Table 1. For implementers, we focused on appropriateness, cost, feasibility, fidelity, penetration, and sustainability. For recipients, we focused on acceptability, adoption, appropriateness, cost, feasibility, and sustainability.

We assessed barriers and facilitators in implementation among implementers (Objective 2) in five domains following the Consolidated Framework for Implementation Research (CFIR) [21]: characteristics of the intervention itself, implementing individuals, inner setting, outer setting, and implementation process. The CFIR proposes that factors in these five domains influence successful implementation.

We assessed barriers and facilitators to using and maintaining the POU arsenic filter among recipients (Objective 3) following the Risks, Attitudes, Norms, Abilities, and Self-Regulation (RANAS) model [22]. The RANAS model proposes that these five factors must be favorable for the successful adoption of a behavior. Definitions for barriers and facilitators at the implementer and household-recipient levels are provided in Table 2.

Both the CFIR and RANAS model have been applied in previous water, sanitation, and hygiene (WaSH) research to describe the factors that influence implementation [24,25,26] and behavior change [27,28,29,30,31,32]. These frameworks are specifically designed to examine these factors for public health interventions. Application of conceptual frameworks allows for more comprehensive evaluation and provides a structure with which to categorize and examine relationships between factors. These frameworks also promote reproducibility and comparison of findings across studies of similar interventions in other contexts, as they provide well-established definitions of constructs [33,34].

**Table 2 ijerph-20-02681-t002:** Barriers and facilitators in implementation of the program and use and maintenance of the POU arsenic filter at the implementer and recipient levels, based on the Consolidated Framework for Implementation Research [21] and Risks, Attitudes, Norms, Abilities, and Self-regulation [22] models, respectively.

Construct	Definition		Examples
**Implementer Level**			
Intervention characteristics	Attributes of the program itself, either as a whole or individual components		Complexity of the program; Relative advantage over comparable interventions
Implementing individuals	Attributes of the personnel implementing the program		Knowledge or beliefs about the POU arsenic filters; Personal commitment to the organization or program
Inner setting	Attributes of the organization implementing the program		Organizational communication systems and norms; Available resources for implementation
Outer setting	Attributes of organizations, policies, and other factors outside the implementing organization		External policy and regulatory environment; Needs and resources of the recipient population
Process	Actions or activities taken to facilitate implementation		Monitoring and learning processes; Engaging local champions to promote the program
**Recipient level**			
Risks	Understanding and awareness of health risks (both perceived and factual knowledge)		Perceived health risks of arsenic; Knowledge and beliefs about arsenic prevalence in groundwater
Attitudes	Beliefs about the costs and benefits of a behavior, feelings towards a particular behavior, health risk, or program		Beliefs about the extent to which the POU arsenic filter can reduce arsenic exposure and mitigate health impacts
Norms	Personal approval/disapproval of a particular behavior, health risk, or program component; perceptions of societal approval/disapproval; and perceptions of to what extent others perform a given behavior		Perceptions of whether others also use the POU arsenic filter; Endorsements from community leaders
Abilities	Capacity, both actual and perceived, to perform the behavior		Perceived and actual ability to use the POU arsenic filter for drinking and cooking, and to perform maintenance on the filter
Self-regulation	Ability to develop plans and routines to facilitate the behavior; cues and feedback to support the behavior		Visual cues to use the POU arsenic filter device; Burden of other household duties that interfere with filter use

### 2.2. Intervention Design and Study Setting

The SHWS program was implemented in the Great Plains region of the United States. The SHWS program was designed through extensive community-centered formative research. A detailed description of the program is published elsewhere [15]. Briefly, the program was implemented by Missouri Breaks Industries Research Incorporated (MBIRI)—a local American Indian-owned research firm—in partnership with local tribal agencies and the Indian Health Service (IHS). Private well users were identified using data provided by the IHS and local municipal tribal water utilities, and MBIRI visited identified private well users to test arsenic levels as part of a community water arsenic testing program. Households exceeding the EPA MCL of 10 μg/L were eligible to receive a POU arsenic filter. A total of 50 households and 84 individuals were enrolled in the trial [16].

Households who enrolled in the study received a Multipure^®^ (Model CB-As-SB, Las Vegas, NV) POU filtration system, which was installed under their kitchen sink/faucet area. The Multipure Drinking Water System is a POU system designed to remove arsenic from water using an absorptive media filter. An absorptive media filter was selected due to concerns voiced by community members during formative research on how reverse osmosis could impact the taste of drinking water [15]. Water meters were included as a separate addition to help households know when to change their filters. Water meters were connected to the POU arsenic filter to monitor water usage and contained a small indicator light on the faucet, which would flash red to indicate when the filter cartridge needed to be replaced (at 960 gallons or approximately one year for average household consumption levels). POU arsenic filters and water meters were installed by plumbers employed by the local tribal housing authority, who also provided households one spare filter cartridge at the time of installation. Included with the spare cartridge was an instruction sheet with diagrams and instructions on how to operate the filter change. Figure 1 shows the filter device.

Households in the intensive study arm only were also provided with cups, water bottles, coffee cups, a large water storage tankard, and a window sticker branded with the study logo designed by an artist in the community. Households were encouraged to place the window sticker near the filter faucet as a visual reminder to use the POU arsenic filter.

Behavior change communication targeted four key behaviors: (1) having one’s well tested for arsenic, (2) having a POU arsenic filter installed, (3) drinking and cooking with arsenic-safe water, and (4) changing the POU arsenic filter cartridge when the water meter flashed red or within 12 months of installation. Behavior change communication was delivered by MBIRI health promoters. Households in both study arms received phone calls at one week and approximately at one, three, and five months after POU arsenic filter installation as part of the SHWS mHealth program. Phone calls delivered the following: (1) information about the health effects of arsenic exposure; (2) reminders to use the filter faucet for all drinking and cooking purposes; (3) reminders to check the indicator light and change the cartridge when the light blinked red; and (4) opportunities to discuss challenges households faced when using their POU arsenic filters and strategies to overcome these challenges.

For households in the intensive arm, in-person visits were conducted at two weeks and one and six months following POU arsenic filter installation. In-person visits were scripted and included videos shown on tablets promoting recommended behaviors targeted as part of the program. Video content included information about the health effects of arsenic, testimonials from community members and the study team, and a demonstration of how to change the filter cartridge. In-person visits also served as an opportunity to discuss challenges households faced when using their POU arsenic filters and strategies to overcome these challenges. In the spring of 2020, in-person visits ceased due to the COVID-19 pandemic. Any remaining in-person visits were converted to mHealth visits, with the same behavior change communication delivered by phone and videos sent by email or Facebook messenger.

### 2.3. Sampling and Recruitment

We contacted every household enrolled in the trial that had not been lost to follow-up at the time of this study (n = 42) to request participation in semi-structured interviews. From each household, we preferentially recruited adults who were responsible for maintaining the POU arsenic filter. When such an individual was unavailable, we recruited any adult over 18 years of age that was enrolled in the study and had been living in the household since the initial installation.

For implementer semi-structured interviews, we sampled the entire team involved in delivery of the behavior change component of the program, and at least one representative from organizations involved in water quality testing and installation of the POU arsenic filters.

### 2.4. Data Collection

We conducted a preliminary phase of semi-structured interviews with seven recipients and four implementers as part of the evaluation of the SHWS program during the RCT. These preliminary interviews were used to provide context for quantitative trial results and explore drivers of program success. These transcripts informed development of the primary interview guides used in this study.

We designed separate guides for recipients and implementers. The recipient guides asked about knowledge and risk perception, attitudes, norms, abilities, and self-regulation (following the RANAS model [22]) for each targeted behavior (i.e., water testing, filter installation, use, and maintenance). The implementer guide asked about implementers’ roles in each component of the program, barriers and facilitators for the target behaviors in different domains (following the CFIR [21]) in delivering each component, and perceptions of success.

Two authors (KE, RS) conducted the preliminary semi-structured interviews in December 2018 and December 2021. A single author (DA) conducted all subsequent interviews in May 2022. A single interviewer conducted each semi-structured interview in participants’ homes or workplaces. Interviews lasted, on average, 45–60 min. We audio-recorded interviews when participants gave permission (94%) and transcribed recordings for analysis. The SHWS RCT was conducted from July 2018 to May 2021.

### 2.5. Data Analysis

We developed an initial codebook of deductive codes for implementation outcomes (Objective 1), constructs from the CFIR for barriers and facilitators for implementers (Objective 2), and constructs from the RANAS model for barriers and facilitators for recipients (Objective 3), based on our conceptual framework.

We then conducted two rounds of codebook testing and revision. First, we applied the codebook to an initial subset of six interviews (four recipients, two implementers). We wrote memos of emergent themes and instances where deductive codes did not fully capture the study context. We used these memos to revise deductive codes and develop inductive codes to reflect emergent themes. Second, we conducted this process of coding, memo writing, and revision again with five additional interviews (four recipients, one implementer) to develop the final codebook.

Once the codebook was finalized, we then proceeded to code the full dataset. All coding was performed using NVIVO (release 1.6.2, QSR International). Four authors (DA, EDT, JG, CMG) participated in codebook development and revision. A single author (DA) conducted all coding once the codebook was finalized.

## 3. Results

### 3.1. Study Sample

We conducted a total of 45 semi-structured interviews with 48 participants. Three interviews were conducted with two participants from the same household simultaneously. For recipients, we conducted 35 interviews across 26 households. Households not included in the sample either were unavailable due to work or other obligations (n = 4), had moved out of the study area or were otherwise unreachable for scheduling (n = 10), or were unwilling to participate in an interview (n = 2). We conducted 13 implementer interviews. Implementers were health promoters and study and intervention supervisors who implemented the behavior change components, and representatives for organizations involved in water testing and installation of the POU arsenic filter. Implementer and recipient demographics are provided in Table 3.

We present results below for perceptions of success, challenges, barriers, and facilitators among implementers, and barriers and facilitators among recipients. A graphical summary of results is depicted in Figure 2.

### 3.2. Perceptions of Success

We assessed successes in terms of the following eight implementation outcomes: acceptability, adoption, appropriateness, cost, feasibility, fidelity, penetration, and sustainability. Implementer and recipient perceptions of each outcome are provided in Table 4. Key trends are described below.

Implementers and recipients described successes most often related to acceptability, appropriateness, and penetration. The SHWS program was widely acceptable to recipients. They described the installation of the POU arsenic filter and health promoter activities as “professional” and “respectful”. Multiple participants described feeling “peace of mind” knowing that the filter was providing a source of arsenic-safe water. Participants had few objections to its design or placement in the kitchen. Both implementers and recipients believed that water quality was a problem in the community that should be addressed, and that the SHWS program was appropriate for community water quality needs.

Implementers reported successes in reaching a high proportion of households in the study area for water arsenic testing. Recipients similarly described that they believed that the study had been successful in raising community awareness. Multiple participants expressed concern that their neighbors may also be exposed to elevated arsenic, and approved of the program’s broad reach:


*“We wanted to be safe and trusted the study—that it was going to help, not only us, but our people. So we’re grateful for you guys being here.”*
(Recipient, mHealth/filter arm).

Implementers perceived that the program had been successful in providing recipients with access to an arsenic-safe water source, as a precursor to adoption. In some cases, implementers believed that the program was successful just by providing a POU arsenic filter, regardless of whether recipients used it. In the words of one implementer:


*“Some things [are] just out of our control. We can’t persuade them but at least give thanks for giving it a try.”*
(Implementer, health promoter).

Recipients reported relatively little impact of health communication encouraging use of the POU arsenic filter on adoption. Where recipients were already concerned about arsenic, they appreciated that the filter provided a convenient arsenic-safe source. However, recipients with low intention to use arsenic-safe water reported relatively little change in this intention after receiving arsenic health communication.

Feasibility, fidelity, cost, and sustainability were challenging. For feasibility, travel to reach households for filter installation and in-person visits for behavior change communication was difficult, in part due to the geographic remoteness of households. Some households took up to three hours one-way to reach by car when driving from the program office, with most taking at least an hour one-way. Furthermore, driving directions were often not readily available on commercial mapping applications (e.g., Google Maps), and implementers thus relied on knowledge of local landmarks to locate households. Contacting households by phone for phone-based behavior change communication was also challenging. However, once households were reached, implementers did not report substantial feasibility concerns.

Implementers reported some challenges in intervention fidelity. Behavior change communication materials as originally scripted reminded participants to check their indicator lights on the water meter as a cue to perform the filter cartridge change. However, due to problems with the short life of the batteries used for the water meters, some water meters stopped functioning prematurely and the batteries had to be changed out, which reset the meter. This meant that the indicator light stopped working for some households. As the implementation team became aware of this problem, they met to discuss this challenge and proposed revised behavior change communication (e.g., using time-based reminders to change the filter 12 months after installation). Fidelity was also challenging with the video component of the intensive arm during the COVID pandemic, when email, text, and Facebook were used to send videos. Some households reported that they did not receive the video messages (e.g., did not routinely check their emails), and implementers reported that it was difficult to monitor whether videos were received and viewed.

Challenges related to cost and sustainability were linked. Recipients appreciated that the installation of the filter device was free but were concerned about the cost of the replacement filters. Many participants reported that the cost of replacement cartridges would be a substantial strain on household budgets. Some described that they would delay or avoid replacing the cartridge due to cost, or would need to make tradeoffs between buying household essentials, such as food, versus replacement cartridges. Some households ordered replacement cartridges online by entering the model name of the POU arsenic filter into a search engine. However, other participants reported that they did not know how they would get replacement cartridges and/or could not afford them, and expected that MBIRI would continue to provide them.

Implementer concerns over cost and sustainability were mixed. Some felt that sustainability was not a concern because households had the filter devices installed and had been provided one replacement cartridge, and that households should and would take responsibility for procuring subsequent replacement cartridges. Other implementers expressed concerns about sustainability and costs, including affordability for low-income households, lack of a local supplier of replacement filter cartridges, and the inability of some households to access filter cartridge suppliers in nearby cities or online (e.g., due to lack of transportation, internet access, or a mailing address capable of receiving deliveries). Implementers noted that some households had old POU arsenic filters that had been installed by previous programs that were no longer in use and needed to be removed before the SHWS filter devices could be installed. Households who had these old systems removed typically reported that they no longer used them because they were unable to afford or obtain replacement filter cartridges.

### 3.3. Barriers and Facilitators among Implementers

We assessed barriers and facilitators among implementers in five categories: intervention characteristics, implementing individuals, inner setting, outer setting, and process, following the CFIR (Table 2). Key trends for each category are described below.

#### 3.3.1. Intervention Characteristics

Implementers who installed the filter devices reported that they were easy to install. Filter devices were provided in prepackaged kits that contained all the necessary fittings to connect with most household plumbing, which streamlined the installation process. Installers also perceived the steel canister that held the POU arsenic filter to be high quality, though they did note that the water meter, separate from the filter (used to indicate when to change the cartridge), had low durability.

Health promoters expressed concerns about the usability of the filter devices, particularly in changing the filter cartridge. Two health promoters reported that they believed the instructions and behavior change communication overestimated how easy changing the filter cartridge was. This finding was corroborated by recipients, who described that the plumbers had told them that the filter change would be “easy” but in practice it was more complex and difficult than they had expected. This is supported by the quantitative findings from the RCT, which found that only 46% of households changed their filter during the two-year study period [35].

#### 3.3.2. Implementing Individuals

Familiarity with the local context and social capital between implementers and community members was an important facilitator, particularly among health promoters. Health promoters were enrolled tribal members and were familiar with the customs and traditions of their tribes. One health promoter was a respected elder who had grown up in the area, had deep social ties with many of the recipients, and spoke the local tribal language fluently. She described how her background made it easier to build rapport with recipients:


*“I know where they come from, and we all grew up the same. And so once you go in there and you’re comfortable with your surroundings, and they’re comfortable with you, I think things are a lot easier…. And there’s a certain amount of respect or standards that we have too, so it’s not like somebody… younger going in there.”*
(Implementer, health promoter).

Whenever younger health promoters encountered difficulty (e.g., households were unwelcoming or unavailable), they sought assistance from this elder implementer who had preexisting relationships with many recipients. This was corroborated by recipients, who described that they felt more comfortable communicating with and trusted implementers who they had known for many years.

However, regardless of age and any pre-existing relationships with community members, communication skills and an outgoing personality were also important. Study and intervention supervisors reported that health promoters with more outgoing, sociable personalities were more effective implementers than those who were shy or timid.

#### 3.3.3. Inner Setting

An important factor was coordination between partners involved in implementation. For some aspects, implementers reported strong coordination that facilitated implementation. For identifying and recruiting households for water quality testing, multiple partners collaborated to share data on the location of private wells, resulting in a comprehensive map of the study area and high coverage of water testing.

However, for other aspects, difficulty in coordination was a barrier. Installations were scheduled at the convenience of the household and households often changed appointments at the last minute. While a health promoter was initially intended to be present during installation, this was difficult to coordinate. To avoid delays in installing the filter devices, health promoter visits were often conducted separately later. A training was held for plumbers to orient them with how to explain the installation of the POU arsenic filters to households and how to explain how to change the filter cartridge. However, not all plumbers were comfortable or had experience communicating with recipients:


*“Away from the reservation you have to have that open communication. And some of the guys, a couple of guys, they don’t have that. They never left the reservation. So the communication was a little bit low.”*
(Implementer, plumber).

When behavior change communication during installation was delivered by plumbers, it was not always consistent. For example, some plumbers emphasized that the filtered water should be used exclusively for drinking, and did not mentioned cooking as they had been instructed to do during their training.

#### 3.3.4. Outer Setting

One important factor in the outer setting was population remoteness and access to communication technology. Many households were difficult to physically access due to poor roads, long travel distances, or both, which was a challenge in filter installation and in-person behavior change communication by health promoters. Health promoters reported that they would sometimes drive two and a half hours to conduct a home visit, only to find that the recipient who had scheduled the visit was not home (e.g., had left to run errands or perform ranching chores). For phone-based behavior change communication, implementers reported challenges contacting recipients, either because phone numbers had been disconnected or recipients did not answer. Recipients corroborated this, with many reporting that they did not recognize phone numbers from the implementation team and never answered calls from unknown numbers, believing them to be spam. Alternative means of contact through voicemails or emails were not always possible, as not all recipients had voicemail setup.

External policies and funding were another important factor, particularly in the sustainability of the POU arsenic filters. Filter installation was funded by the IHS. However, under IHS regulations, funding could not be used for subsequent operations and maintenance. Implementers from IHS reported that they encouraged other implementers to consider how households would access a sustainable supply of replacement filter cartridges, but were unable to take further action. Among other implementers, there was disagreement regarding who should be responsible for ensuring access to a sustainable supply of filter cartridges. Some believed this should be the responsibility of the homeowner. Others believed that homeowners could not or would not pay for replacement cartridges themselves and were concerned that no local business stocked the cartridges at an affordable price. Ultimately, responsibilities for ensuring a sustainable filter cartridge supply were not clearly established, with implementers from stakeholder groups each reporting that others should be responsible for ensuring a sustainable supply.

#### 3.3.5. Process

Identifying and adapting to challenges was an important factor. The implementation team held routine meetings to discuss progress and identify challenges. Health promoters had the flexibility to adapt to challenges and develop solutions. When in-person visits stopped in response to the COVID-19 pandemic, the implementation team adapted to develop video-based behavior change communication as a replacement, which was perceived as successful by both implementers and recipients. While the implementation team effectively identified challenges and developed solutions, ensuring that households consistently received adapted materials was sometimes challenging.

### 3.4. Barriers and Facilitators among Recipients

We assessed barriers and facilitators among recipients in five categories: risks, attitudes, norms, abilities, and self-regulation, following the RANAS model (Table 2). Key trends for each category are described below. We found few differences in barriers and facilitators between the two study arms, and therefore report the findings for both arms together.

#### 3.4.1. Risks

Individuals who were more knowledgeable about arsenic and its health effects were more likely to believe that the consistent use of arsenic-safe water was important. Knowledgeable recipients often had learned about arsenic before the program, for example through employment or post-secondary education. For those without prior knowledge, recipients reported that the program raised awareness of dangerous arsenic levels in their water. However, many wanted more information on the specific health effects of arsenic, beyond the health communication already provided as part of the SHWS program which discussed the association between arsenic in water and cancers, heart disease, and diabetes. Some recipients sought information elsewhere but found interpreting complex toxicologic or medical information challenging:


*“It was kind of a lot to try reading. I’m just overwhelmed with Google searches... There’s a lot of stuff to scroll through and try to find what would be more informative.”*
(Recipient, mHealth/filter arm).

In some cases, prior arsenic knowledge was a barrier. Some recipients were familiar with the previous EPA MCL for arsenic of 50 μg/L (applicable before 2001, when the current MCL of 10 μg/L was set). In these cases, individuals often believed that the new level was unnecessary or overly cautious.

Personal and family health history was also an important factor. Individuals who had personally experienced health effects perceived to be related to arsenic (e.g., cancer, stroke, premature death) believed the use of arsenic-safe water was important. Conversely, individuals who had grown up in the community drinking the water with no perceived health effects were less likely to believe that arsenic-safe water was important. In some cases, people reported generations of family members who drank the same water with no perceived effects, strengthening the perception that arsenic was not a major risk:


*I wasn’t very concerned because like I said, we’ve lived here all this time, and no one’s died of arsenic poisoning that I know of yet.”*
(Recipient, intensive arm).

#### 3.4.2. Attitudes

Attitudes were an important factor in behaviors related to using the POU arsenic filter, particularly palatability of the filtered water and trust that the filter would effectively remove arsenic from the water and protect against health effects.

The poor palatability of water was a barrier. A common complaint was that the filtered water was not as cold, or, less commonly, that filtering changed the taste. For most people who were strongly concerned about health effects, these preferences were not enough to prevent the use of the filter, though some opted to purchase bottled water instead. However, for individuals who were unconcerned or skeptical of the health effects, low palatability discouraged filter use, and these participants rarely used an alternative arsenic-safe source.

Skepticism of the filter efficacy was another barrier. Some individuals believed that arsenic was present and had negative health effects but were skeptical that the filter would remove it. Some of these individuals used other arsenic-safe water sources (e.g., bottled water), but others continued using unfiltered well water. Skepticism was influenced on two levels. First, the indicator light used to indicate the filter needed to be changed (which was connected to a water meter and was a separate component from the filter itself) flashed red if the filter needed to be changed or lit up solid red if the battery was low. The use of the same color for two very different signals caused confusion among participants. For example, one participant reported changing her filter mistakenly because her indicator light was solid red (low battery) and she thought her filter was not working. Second, loose or leaking filter faucets, which occurred in two households and were subsequently fixed.

More distally, skepticism was also influenced by historical experiences with poor quality programming and exploitation, particularly by government agencies. Multiple participants thought that the SHWS program was a government program because of partnerships with tribal authorities and the IHS, and had pre-existing perceptions of government programs as being low quality. For example, a participant described that she did not report problems with the filter device because she did not believe that anyone from the implementation team would respond to fix them.

Skepticism and mistrust were mitigated by several activities. The SHWS program included sampling the filter faucet (treated) and the kitchen faucet (untreated) upon filter installation, and mailing households test results comparing the levels to indicate filter effectiveness. The delivery of this information to the recipients helped build confidence that the filter was working and encouraged consistent use:


*“I guess I learned a long time ago, you kind of have to pay attention to what’s in black and white, because not always is the word truthful. So we did see the results in black and white, and that gave us some assurance.”*
(Recipient, mHealth/filter arm).

The implementation team demonstrated drinking a glass of water from the filter faucet during household visits to show recipients that the filtered water was safe to build trust. Additionally, for recipients in the intensive arm, videos showing testimonials and endorsements from community members and elders also helped build trust.

Some individuals believed that arsenic was a problem and that the filter could effectively remove it, but that it was too late for them to mitigate any health impacts. These individuals were often older and had personally experienced or observed others with a high burden of other competing hardships, such as exposure to substance abuse or violence. When asked whether she used the filter, one recipient replied, “why bother”, and noted that friends and family had died from cancer, drug addiction, and gun violence:


*“I’m getting old and am going to die soon anyway… I try not to think about it. When I die, I die.”*
(Recipient, intensive arm).

However, this same recipient was hopeful that future generations would experience a better life and encouraged her teenage grandchildren living in the home to use arsenic-safe water. This attitude was shared by many recipients with children, where older adults were more diligent about encouraging filter use among younger children than using it themselves.

#### 3.4.3. Norms

Norms were not a strong barrier or facilitator. Recipients reported that they did not typically discuss water with others outside the home. We commonly found that filter users and non-users lived together within households and did not routinely attempt to influence each other’s behavior. One recipient described briefly encouraging her teenage children to drink from the filter faucet but ultimately respecting their autonomy:


*“Some people, if they don’t want to use it, they won’t, no matter how much you convince them. You can suggest it… But if they’re not going to use it, that’s not my problem.”*
(Recipient, intensive arm).

#### 3.4.4. Ability

Ability was an important barrier in maintaining the filter (i.e., replacing the cartridge when it reached capacity). We identified two key factors: information and physical capacity.

Informational capacity related to understanding how to disassemble the filter cannister and replace the cartridge. Some but not all recipients reported that they had received a demonstration from the plumbers who installed the filter. Information capacity was hindered by lack of information sharing between household members. Working adults were often responsible for changing the filter but may not have been home at the time the filter was installed. Multiple recipients reported that their family members had not relayed verbal instructions from in-person visits or phone calls, and that printed materials (e.g., owner’s manual for the filter device) had been lost. The intensive arm included a video from a community member changing the filter cartridge which was showed in person and through Facebook and text messages.

Informational capacity varied by gender and age. Men reported more experience with day-to-day home repair or employment using mechanical skills (e.g., auto repair), which gave them the confidence and skills to maintain the filter. Women were less confident and less willing to attempt maintenance without instructions. Younger recipients regardless of gender were more likely to look up information, for example, by using an online search engine to look up the model name and watch instructional videos that were not part of the study materials (e.g., publicly available videos on YouTube for similar filter devices). Across all recipients, individuals with low confidence expressed concern that attempting to replace the filter cartridge might interfere with the water supply throughout the house, and they preferred to continue using the expired filter rather than risk losing water access.

Physical capacity related to the ability to access the filter cannister under the sink and maneuver it to replace the filter cartridge. Individuals with limited mobility, particularly older individuals or those with physical disabilities, reported having difficulty reaching under the sink. Even individuals with adequate mobility sometimes lacked the strength to loosen the mounting hardware or lift the cannister to change the filter cartridges, particularly as cartridges became heavy once saturated with water.

For both informational and physical capacity, barriers were mitigated by having a strong social support system. Several individuals who lacked capacity themselves asked friends, family, or neighbors to assist them. Individuals who lacked capacity themselves and who also did not have a support network often identified that maintenance was necessary but continued to use the expired filter cartridge or reverted back to the unfiltered faucet.

#### 3.4.5. Self-Regulation

For behaviors related to using the filter, most recipients did not identify any specific facilitators besides becoming accustomed to the filter over time. For many participants, acclimating to the new filter was challenging and took weeks or months. During the adjustment period, recipients reported lapses where they used the unfiltered water. The kitchen faucet was retained in households to try to preserve the life of the filter cartridge for as long as possible—using the filter faucet for washing dishes and hands would shorten the life of the filter—but some participants described that having multiple faucets for different uses (e.g., using the filtered faucet for cooking and drinking but the unfiltered faucet for washing dishes and hands) was complicated and hindered habit formation. For some households with young children, the novelty of the filter was a facilitator, where children were excited to try the new device:


*“The kids, I think they got used to it right away because the little ones, they just jump up there, and just turn it on and get their water.”*
(Recipient, mHealth/filter arm).

Visual cues to action were important. For filter use, participants who placed the window sticker with the study logo above the kitchen sink reported that it was a helpful reminder to use the filter. For filter maintenance, households relied on the indicator light to know when to perform the filter change. As mentioned earlier, a solid red light indicated that the water meter batteries were low, while a blinking red light indicated that the filter cartridge had reached capacity. However, there was confusion among recipients about this, with some recipients reporting changing their filter cartridge just several weeks after installation in response to the low battery indicator. When this issue was identified, health promoters then recommended changing filter cartridges after 12 months (time-based indicator) rather than relying on the indicator lights.

Some recipients believed that there would be a visual sign to change an expired cartridge (e.g., a slow down or complete stop to the water flow or a change in the water appearance), and that any filter still delivering water at the usual flow rate was functioning as designed and was safe to use. When behavior change communication asked if recipients were having any problems, many interpreted this to mean major interruptions or malfunctions in the filter flow, and did not report or seek help for smaller issues such as indicator light malfunctions or problems with the cartridge change:


*“Well, I’ve never had a problem with the unit. Well, I guess I did have a, you know, just minor things. A lot of times the batteries come loose or just pop out [from water meter]... The filter itself never had any flow problems that was fine. Still has the same flow that it did when we first installed it. So I had never had any problem with it giving me problems of not delivering water or anything like that.”*
(Recipient, intensive arm).

The timing of behavior change communication was also a barrier, particularly in relation to filter maintenance. During the first six months, when most home visits and phone calls were made, recipients experienced few problems with the filters and were not yet ready to perform the cartridge change. When recipients were ready to perform the filter change, some had forgotten earlier communication from health promoters.

## 4. Discussion

This study evaluated the implementation of the community-led SHWS program, the first RCT designed to mitigate arsenic exposure among private well users in American Indian communities. We assessed program successes and challenges using a framework of implementation outcomes [23]. We assessed facilitators and barriers at the implementer and recipient levels, following the CFIR [21] and RANAS model [22], respectively. At the implementer level, facilitators were the rapport and trust built between individual implementation staff members and participating households. Barriers included remoteness of households, coordinating implementers’ and households’ availability for the installation of the POU arsenic filters, and difficulty securing a local supplier for replacement filter cartridges. At the recipient level, facilitators included knowledge of the health risks of arsenic, perceived effectiveness of the filter, and visual cues provided as part of the program to promote habit formation. Barriers included attitudes towards water taste and temperature and the inability to procure or install replacement filter cartridges.

These findings complement those from our RCT of the SHWS program, which found that the program significantly reduced arsenic exposure and increased exclusive use of arsenic-safe water [16]. Delivery of the community-led SHWS program resulted in a 48% reduction in urinary arsenic (biomarker of arsenic exposure) from baseline to the final follow-up visit during the 2-year study period. Through this study, we were able to identify important barriers and facilitators to the delivery of the SHWS arsenic mitigation program at the recipient and implementer levels, which should be targeted during program scaling.

### 4.1. Implementation Successes and Associated Barriers and Facilitators

Key SHWS program successes were the acceptability of the intervention among recipients, appropriateness of the program to the community’s needs and the cultural context, and penetration in terms of reaching a high proportion of private well users in the target population. These successes were facilitated by the characteristics of implementing individuals and organizations, and the process of engaging key stakeholders from the community and local agencies for program design and delivery (i.e., CFIR domains for implementing individuals, inner setting, and process [21]).

Implementers noted that the program had been designed by tribal members [15], which contributed to its acceptability and appropriateness. Recipients reported concerns about water contamination and noted that the program addressed an important community need. Notably, concerns were not restricted to arsenic but rather reflected a broad range of water contaminants, including uranium, agricultural runoff, and mining-related toxicants. These concerns have been substantiated in other studies indicating high prevalence of these water contaminants [11,12,36,37]. While the SHWS program did not address these non-arsenic contaminants, our study suggests that broad concern for water quality and contamination can increase the acceptability of mitigation programs, even if the filtration technology does not specifically target all the contaminants of concern.

SHWS program successes were also facilitated by the characteristics of the implementation team. Health promoters were enrolled tribal members, who were familiar with the community context and had lived in the community for years, building social ties with enrolled households. These social ties helped build trust in the program and the POU arsenic filter’s effectiveness, improving acceptability and appropriateness. Social ties also helped implementers identify eligible households within the community and facilitated follow up with households after enrollment, improving the penetration of the program in the target community.

Social ties and community trust have been found to be important drivers of WaSH intervention success in low- and middle-income countries [24,38], and this research suggests that they are equally important in our partner American Indian communities. Trust and social ties are likely particularly relevant for American Indian communities, which have a long history of exploitation and marginalization. Tribes have been forcibly relocated off their ancestral homelands, often onto lands that were historically considered marginal or less valuable [39]. Tribal lands and natural resources have been repeatedly contaminated and exploited—both historically and present-day—by federal and state governments and non-tribal businesses without tribal consent, often on land that tribes consider sacred [39,40,41,42].

Renumeration for these abuses to date has been severely lacking [39,40,41,42]. Counties that are home to American Indian Nations continue to be among the poorest in North America, with higher incidence of chronic conditions such as heart disease and diabetes [43,44,45]. Infrastructure on tribal lands remains under-developed compared to neighboring non-tribal areas [46,47].

This historical trauma is reflected in the current experience of American Indian communities and continues to influence program delivery. Mistrust among American Indian communities of government programming has been well-documented [48,49]. Participants in our study discussed how they had received low quality programming unrelated to the SHWS program (e.g., housing assistance programs) that made them distrustful of other programming, particularly when they perceived it to be delivered by government agencies. In this study, we found that personal relationships with SHWS program implementers helped mitigate this mistrust, as did engagement of community members in the program design process to ensure the program met community needs. As such, similar WaSH programs delivered in American Indian communities should meaningfully engage community members and trusted community leaders (e.g., tribal elders) in the design phase, and the implementation teams should be comprised of tribal members.

Participants who were concerned about arsenic and believed the arsenic filter device would effectively remove arsenic used the filter consistently. Similarly, participants who were unconcerned about arsenic or did not believe that the arsenic filter device would effectively remove arsenic did not report using the filter device, nor did they report that the health communication meaningfully influenced their attitudes or practices. It should be noted that the health communication provided was only one aspect of the SHWS program delivery. It is possible that the community promoters calling and making in-person visits to check in on households was more important than the actual content of the health communication itself, given that community-led program implementation was important in building trust in the program.

In the SHWS RCT, we assessed the impact of the SHWS program on behavioral determinants of the exclusive use of arsenic-safe water using a Likert scale questionnaire. We found that the SHWS program significantly increased perceived vulnerability of use of arsenic unsafe water and self-efficacy to reduce ones arsenic exposure, and that these psychosocial factors were associated with higher exclusive use of arsenic-safe water at follow-up [50]. However, the SHWS program did not significantly change arsenic knowledge, nor was arsenic knowledge associated with exclusive use of arsenic-safe water. Our current qualitative evaluation yielded consistent findings, with abilities (e.g., knowledge and/or support systems to change the arsenic filter cartridge) and attitudes (e.g., beliefs that the filter could effectively remove arsenic and concern for the health implications of arsenic exposure) identified as key behavioral determinants of use of the arsenic filter, and arsenic awareness being unchanged by the program. Norms (e.g., encouragement from family members to use the filter) were also not found to meaningfully drive behaviors

### 4.2. Implementation Challenges and Associated Barriers and Facilitators

We found few differences in facilitators and barriers between the two study arms. This may be in part because the filter device was the same across both arms, as was the mHealth component, and most of the facilitators and barriers described by implementers and participants related to these two components. This may also reflect fidelity challenges associated with adapting and delivering the in-person components (e.g., videos) through mHealth and Facebook during the COVID-19 pandemic, where ensuring that households received and viewed video content was more challenging, and some households may not have viewed these materials as intended.

Sustainability and costs of the POU arsenic filters were key challenges. These challenges were linked, as costs and availability of replacement cartridges were a barrier to sustained use and maintenance. Households were provided with one spare filter cartridge but afterwards were expected to procure their own. While some households ordered replacement arsenic cartridges online, other recipients expressed concern that there was no locally available supply of affordable arsenic filter cartridges. These concerns are substantiated by the fact that plumbers who installed the point-of-use arsenic filters for the SHWS program reported that sometimes they had to remove old, no longer used filter systems before installing the SHWS POU arsenic filter. Recipients corroborated this by reporting that some had previously received filter systems in the early 2000’s but stopped using them, in part because they were unable to maintain a supply of replacement cartridges.

Policy and funding structures at the implementer level were barriers to a sustainable and affordable supply of arsenic filter cartridges. The installation of the filter devices for this program was funded by the IHS. However, federal regulations dictate that IHS funding can be used to establish water services but not to subsequently operate and maintain them. While some implementers were concerned about sustainability, these policies limited the actions they could take. Other implementers perceived the SHWS program to be a temporary solution, with the long-term goal of connecting households to the municipal water supply. However, formative research for the SHWS program indicated that some households may be reluctant to connect to the municipal supply, in part because of preferences regarding the temperature and taste of drinking water and the potential cost and perceived reliability of the municipal water supply [15].

Beyond household preferences, policy and funding structures complicate connecting households to the municipal water supply as the long-term solution. The municipal water supply project serving the SHWS program area was designed in the 1980s and funded by the federal government [51]. At the time of construction, some areas opted out of being served by water lines. In areas that were not included under the original scope of work, tribal authorities are now responsible for funding the construction and maintenance of any further water lines. Furthermore, even in areas served by the original network, federal funds that were used for construction have expired [52].

Programs for new household municipal water supply connections are limited, and the costs of connections are prohibitive for most households without external financial support. As such, the expectation among implementers that households should be responsible for maintaining their own supply of arsenic filter cartridges, or that they should connect to the municipal network as the preferred alternative, may not be realistic. The SHWS program was designed to provide households with access to arsenic-safe water and provide information on how to use and maintain arsenic filter systems. The program was not designed to address supply chain or affordability challenges in sourcing replacement arsenic filter cartridges in the long-term. However, our findings suggest that additional interventions at the policy level are also needed to ensure that the SHWS program can be sustained.

Independent of costs and arsenic filter supply challenges, sustainability was also a challenge in terms of households’ ability to change the filter cartridge when needed. These challenges were divided into two categories: information about when and how to change the cartridge and physical ability to change the filter cartridge. For informational challenges, cues to action were an important behavioral determinant. In households where the indicators light on the base of the filter faucet functioned, this served as a visual cue to replace the arsenic filter cartridge. However, in some households after a few months of installation this indicator light became solid red light indicating a low battery, and when batteries were changed, the water meter reading was reset to 0. When these challenges were detected, implementers used time-based reminders for households, recommending that households change their filter cartridge after 12 months. mHealth communication was adapted to include the time when the filter cartridge needed to be changed, and a video with a community member showing how to change the filter cartridge was sent by text message and Facebook. Knowledge on how to change the filter cartridge and the physical ability to change cartridges were also barriers, particularly for older populations and those with physical limitations. Future arsenic mitigation programs need to target these important challenges faced by households in changing their arsenic filter cartridges.

We recommend adapting future arsenic mitigation programs to encourage strengthening social support networks to assist with changing the filter cartridge when needed. For households where all members face challenges with their physical abilities and no one in the household can be readily identified to perform the filter cartridge change, communication encouraging individuals to identify and develop social support systems (e.g., identifying and contacting a family member or neighbor to provide assistance) may be more effective. Other studies have found that social support is an important facilitator for other WaSH behaviors [24,31,38], and our findings corroborate those results.

Ultimately, behavior change communication for the SHWS program may need to be tailored to specific groups, as the type of support required by different individuals varies by demographic. For example, we found that the elderly and individuals with reduced mobility required more support for physical abilities when changing arsenic filter cartridges, indicating that they may benefit from more information support. Similarly, younger recipients were more comfortable receiving video messages by text and Facebook and using phone voicemail systems, while older recipients often preferred postal mail and face-to-face communication. Further research to tailor implementation strategies by demographic group may improve future program effectiveness.

Filter selection was driven by community preferences on the aesthetic quality of water and by local hydrogeology and the technical ability of the filter to efficiently remove target contaminants [11]. Quantitative results from the SHWS RCT indicated that the POU arsenic filters met and exceeded expectations, with 93% of filter-proceeding water below the arsenic MCL at the final follow-up visit during the 2-year study period [35]. This finding was observed despite only 46% of households reporting changing their arsenic filter cartridge within the 2-year program period. Given the 48% reduction in urinary arsenic observed for program recipients from baseline to the final follow-up visit during the 2-year study period (indicating high use of arsenic-safe water) [16], these findings suggests that the POU arsenic filter cartridges installed can effectively reduce arsenic for a much longer duration than originally estimated. Future studies are needed to evaluate the gallons of filtered water that can be produced before arsenic filter failure in the hydrogeological conditions in our study setting, and the recommendations for the filter life need to be revised accordingly.

### 4.3. Strengths and Limitations

This study has several strengths. First, we leveraged the CFIR and RANAS model a priori in designing data collection tools and in the analysis stage, to more comprehensively capture the range of barriers and facilitators relevant to implementing the SHWS program and using the POU arsenic filters. Furthermore, we sampled a large proportion of the households enrolled in the trial, and we achieved saturation across many of the themes reported in this study. Second, this study was conducted approximately four years after the start of SHWS program implementation, allowing us to identify a wider range of barriers and facilitators over time, particularly related to long-term operation and maintenance of the intervention at the household level. Third, there are very few qualitative evaluations of arsenic mitigation programs. Most arsenic mitigation studies are cross-sectional and focus on quantitative data collection methods [53,54,55,56,57,58,59,60,61]. Through conducting this evaluation, we were able to identify the facilitators and barriers in the implementation of the SHWS program at the recipient and the implementer levels, allowing us to identify areas for further program refinement before scaling.

We identified two main limitations for this study. First, we experienced challenges with participant recall of activities. Because two or more years had passed since initial POU arsenic filter installation and early behavior change communication activities, some participants did not recall receiving certain components of the intervention (e.g., specific phone calls or home visits). Second, while we sampled a large portion of the households not lost to follow-up, households with the most severe challenges may be more likely to be lost to follow-up or not enrolled at all. For example, households without electricity would not have been able to have the arsenic filter installed and thus were not eligible to be enrolled, and similarly, households who had their electricity disconnected during the study would not have been able to continue to operate the filter device and may have shifted their home due to financial insecurity. As such, this study may underestimate challenges, particularly for the most vulnerable households.

## 5. Conclusions

The community-led SHWS is the first RCT of a program designed to mitigate arsenic exposure through groundwater among private well users in American Indian communities. The program significantly reduced arsenic exposure, as measured by urinary arsenic concentration [16]. We attribute this success to the high acceptability, appropriateness, and penetration of the community-led SHWS program among the target population. These successes were facilitated by collaborations with tribal agencies, engagement of community members in the program design and implementation, and the implementation team’s social ties to communities as enrolled tribal members. Trust was an important facilitator for acceptability and appropriateness. Lack of trust was also a barrier to acceptability, driven in part by prior experiences with low quality programming and the exploitation of American Indian communities, which was mitigated by social ties between SHWS implementers and recipients.

Sustainability and costs were challenging. Recipients reported concerns over their ability to maintain their POU arsenic filters over time, particularly with regard to the lack of a local, affordable supply of filter cartridges. Addressing these challenges was complicated by policy and funding structures, as the IHS funded installation of filters, and federal regulations prohibit using IHS funds for the purchasing of replacement filter cartridges over time. Other important barriers to sustainability included physical mobility and strength challenges, as well as low self-efficacy that hindered changing the POU arsenic filter cartridges.

Future arsenic mitigation programs conducted in partnership with American Indian communities should meaningfully engage community members in the program design and implementation to strengthen acceptability, appropriateness, and penetration. Adaptions to strengthen social support for households with reduced ability for maintaining the arsenic filter device may improve the success of program implementation. We also recommend that future arsenic mitigation programs take steps to address challenges related to ensuring an affordable, sustainable supply of arsenic filter cartridges to support long-term use and maintenance, which may require interventions at the organizational or policy level to address relevant barriers.

## Figures and Tables

**Figure 1 ijerph-20-02681-f001:**
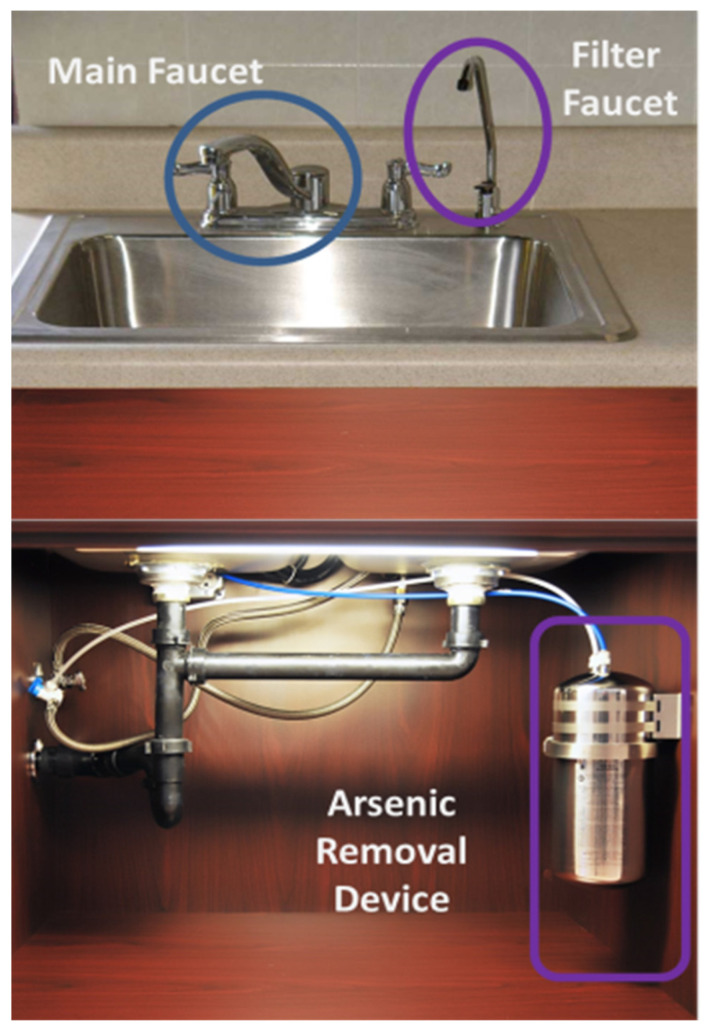
Filter device, showing POU arsenic filter installed under kitchen sinks and filter faucet.

**Figure 2 ijerph-20-02681-f002:**
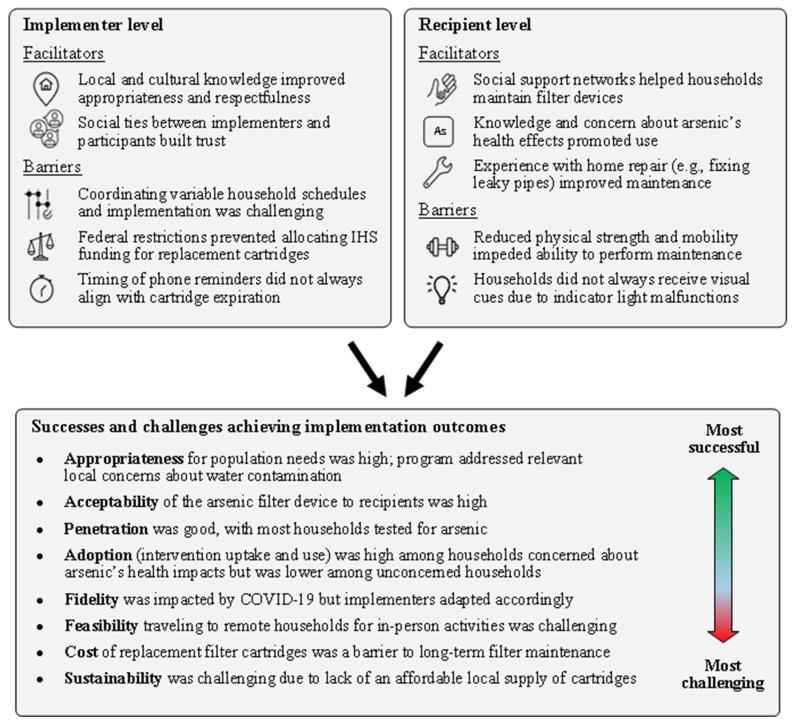
Graphical summary of study results.

**Table 1 ijerph-20-02681-t001:** Implementation outcomes as measures of perceived program success.

Outcome	Definition
Acceptability	Perception that the program, or its component parts, is agreeable and satisfactory
Adoption	Intention, decision, or action to uptake key behaviors for the program (e.g., obtaining an arsenic test, using arsenic-safe water)
Appropriateness	Perceived fit, relevance, or compatibility of the program with the context, or perceived fit of the intervention to address locally relevant issues or problems
Cost	Cost of the program and implementation efforts
Feasibility	The extent to which the program can be successfully delivered and/or used within with local context
Fidelity	The extent to which the program is delivered as originally developed and specified in program plans and protocols
Penetration	Coverage area and intensity of exposure to the program among the target population
Sustainability	The extent to which the POU arsenic filters and associated key behaviors are maintained within the target population

**Table 3 ijerph-20-02681-t003:** Demographic traits of the recipient and implementer samples.

Demographic Trait	Number of Participants
**Recipients (n = 35)**	
Age	
18–35	3
36–50	7
51–65	7
66+	18
Gender	
Male	17
Female	18
Intervention arm	
mHealth/filter	23
Intensive	12
Self-reported filter use †	
Exclusive user	8
Partial user	26
Non-user	1
**Implementers (n = 13)**	
Behavior change implementers	
Health promoters	6
Study and intervention supervisors	2
Role of partner organizations	
Water quality testing	3
Filter installation	1
Intervention design and funding	1

† We defined self-reported filter use categories as: Exclusive user (self-reported exclusive use of arsenic filter for drinking and cooking at the final follow-up visit); Partial user (partial use of the arsenic filter for either drinking or cooking at the final follow-up visit); Non-user (no use of arsenic filter for drinking nor cooking at the final follow-up visit).

**Table 4 ijerph-20-02681-t004:** Implementer and recipient perceptions of intervention successes and challenges.

Outcome	Implementer Perceptions		Recipient Perceptions
Acceptability	Assessed at recipient level only.		Most recipients liked the POU arsenic filter. They described the installation as “professional” and “respectful”. Recipients reported that the filter gave them “peace of mind” for arsenic-safe water. Some recipients reported low acceptability associated with installation problems (e.g., filter faucet improperly secured on the kitchen sink leading to wobbling).
Adoption	Implementers perceived that increased arsenic awareness and opportunities to access arsenic-safe water were successes. Some implementers reported that they had achieved success just by installing the filter, and whether recipients decided to use it was outside of their control. Others perceived that use of the filter device by even some recipients was a success, believing that some recipients would never change their behavior.		POU arsenic filters provided recipients that wanted to use arsenic-safe water with an opportunity to do so. Some recipients were concerned about arsenic in their well water but did not believe the filter device would effectively remove it. These individuals typically used bottled water. For recipients that were unconcerned about arsenic and did not intend to switch to arsenic-safe water, health communication did little to change their attitudes or practices.
Appropriateness	Implementers believed that arsenic exposure was a locally relevant problem. Implementing organizations and agencies had missions and mandates to provide safe water and/or protect health, and the program was perceived to align well with those needs. Implementers perceived the program to be culturally competent because it was designed with input from community and tribal members, and many implementers were tribal members who were aware of community needs.		Many recipients were concerned about water contamination and associated health impacts from arsenic and other toxicants (e.g., solid waste dumping, farming and mining chemicals). Recipients believed that the filter addressed a relevant community need and that water quality testing and raising awareness about arsenic in the community was important.
Cost	Some implementers were concerned about the affordability of replacement filter cartridges for low-income households, but otherwise cost was not a major concern mentioned by implementers.		Recipients appreciated that the filter was installed at no cost. Many expressed concerned over the high cost of replacement cartridges. Some reported that they may need to make cuts in other areas of the household budget to afford one, or that they would delay replacing an expired cartridge to save money.
Fidelity	The short life of batteries in the water meter necessitated adaptation to communication regarding using the indicator light to signal filter cartridge changes. The COVID-19 pandemic also disrupted implementation and necessitated adaptation of in-person components to remote delivery.		Recipient reports confirmed implementer descriptions of some variation within study arms about the content and format of behavior change communication.
Feasibility	Feasibility challenges centered around travel to remote households over poor quality roads for the installation of POU arsenic filters and in-person visits. Once at the household, implementers described the filter device cannister as well-designed and easy to install.		Some households reported initial difficulty changing habits to routinely use the filter faucet for drinking and cooking, but in the long-term reported few difficulties. Feasibility of maintenance was more challenging, with many households reporting trouble due to either lack of physical ability or sufficient information to change the filter cartridge.
Penetration	Implementers perceived penetration to be good in terms of conducting water testing for a very high proportion of private well users in the target area, even those in remote or difficult to access areas due to poor road quality. Implementers reported challenges in penetration of behavior change communication with enrolled households (e.g., households not answering the phone).		Recipients corroborated difficulties receiving behavior change communication by phone. Many reported that they did not recognize phone numbers of the study team and believed they were spam calls. Study protocols dictated that health promoters should leave a voicemail message, but many recipients, particularly the elderly, did not have a voicemail setup.
Sustainability	Mixed views on sustainability. Some implementers perceived that since the filter devices had been installed by the program, households could and should be responsible for maintaining them. Others believed that, since an affordable option for replacement cartridges was not readily available locally, the intervention would not be sustained.		Recipients expressed concerns over their ability to locally source and afford filter cartridges in the long-term. Many expected that the study would continue to provide them, and had no alternative plan to source filters when they were no longer available for free through the study.

## Data Availability

The participants of this study did not give written consent for their data to be shared publicly, so due to the sensitive nature of the research, supporting data is not available.

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
