# Peer review of "Implementing a Community-Led Arsenic Mitigation Intervention for Private Well Users in American Indian Communities: A Qualitative Evaluation of the Strong Heart Water Study Program"

_ijerph, 2023, doi:10.3390/ijerph20032681_

Round 1
Reviewer 1 Report
This study offers new insights on the successes and challenges of implementing an arsenic mitigation program tailored to American Indian households, which can inform future programs in partnership with these and potentially other similarly affected communities. Accept after minor revision.
Comments:
(1) These findings and assessment results can be shown more intuitively by using figures.
(2) When using assessment model, it should first be introduced, and then specifically explain how the model is applied to this study.
Author Response
Reviewer 1
|
(1) These findings and assessment results can be shown more intuitively by using figures. |
We thank the reviewers for their thoughtful consideration of our manuscript. Please see addition of figure 1. |
|
(2) When using assessment model, it should first be introduced, and then specifically explain how the model is applied to this study. |
We have added text at the start of the methods section to state which frameworks we will be applying: In this study, we applied qualitative methods, using semi-structured interviews with implementers and recipients. We apply conceptual frameworks from implementation and behavioral science [21–23] to evaluate successes, barriers, and facilitators to implementation and intervention uptake. Specific methods are described below.
We have then later added additional text to explain how the CFIR and RANAS models have been applied in this study, and how they strengthen our analysis:
Both the CFIR and RANAS model have been applied in previous water, sanitation, and hygiene (WaSH) research to describe the factors that influence implementation [24–26] and behavior change [27–32]. These frameworks are specifically designed to examine these factors for public health interventions. Application of conceptual frameworks allows for more comprehensive evaluation and provides a structure with which to categorize and examine relationships between factors. These frameworks also promote reproducibility and comparison of findings across studies of similar interventions in other contexts, as they provide well established definitions of constructs [33,34].
|

Reviewer 2 Report
Hi,
The research highlights the practical challenges and limitations in resolving access and availability of safe drinking water not only to Indian communities but also to other communities around the world. I enjoyed reading your research article. Please see Attached file for a minor editorial change.
Here are some comments on my part:
*Would it be possible to provide the reader some geographical background as it pertains to the study area (i.e., a map for example!)
*In the study, the authors mentioned the remoteness of site as a real challenge. Indeed, I have experienced such challenge in some drinking water quality study that I have conducted overseas. In order for the reader to better conceptualized this issue, can the author indicate the typical distance between households (i.e, on a radial distance for example!)
*I was wondering if motivation would be a standalone factor among the eight outcomes described in Section 3.2?
Line 428: ….., and (therefor) therefore report the findings for
Author Response
Reviewer 2
|
*Would it be possible to provide the reader some geographical background as it pertains to the study area (i.e., a map for example!) |
We thank the reviewers for their thoughtful consideration of our manuscript. We have not added a map specifically depicting the study area to protect the confidentiality of participants in this study. There are a limited number of American Indian reservations and communities in the Great Plains region, and communities are small. Given other contextual details reported in the study, we are unwilling to risk accidentally deidentifying participants by providing greater specificity of the location. |
|
*In the study, the authors mentioned the remoteness of site as a real challenge. Indeed, I have experienced such challenge in some drinking water quality study that I have conducted overseas. In order for the reader to better conceptualized this issue, can the author indicate the typical distance between households (i.e, on a radial distance for example!) |
We have added a description of the travel time to reach households from the headquarters of the implementing organization:
Some households took up to three hours one-way to reach by car when driving from the implementation headquarters, with most taking at least an hour one-way. Furthermore, driving directions were often not readily available on commercial mapping applications (e.g., Google Maps), and implementers thus relied on knowledge of local landmarks to locate households.
We have quantified distance in terms of driving time. Because of the poor quality of roads and limited availability of roads, radial distance may not be meaningful (e.g., there are no roads that closely follow the most direct point-to-point line and implementers must drive 1-2 hours in the other direction to access passable roads). |
|
*I was wondering if motivation would be a standalone factor among the eight outcomes described in Section 3.2? |
Thank you for this comment. Section 3.2 contains information on implementation outcomes. Implementation outcomes are the intermediate results of program activities, that are essential to achieving long-term health impacts. For example, do people use the intervention (adoption), is the intervention delivered as intended (fidelity). Motivation is a facilitator to achieving these outcomes, not an outcome itself. This is why we have included it in section 3.3 for barriers/facilitators among implementers. We have added a figure that illustrates the relationships between all constructs in the study, which should help clarify this distinction for readers. |
|
Line 428: ….., and (therefor) therefore report the findings for |
We have corrected this typo. |

Reviewer 3 Report
The work still need a lot of improvements within different levels from title to results, please find below some comments that may improve the work:
Title:
Too long need to be changed and make it shorter and keep representing the core value of the work.
Abstract:
The abstract should be rephrased: 1) including, in a nutshell, the hypothesis, objective, the key methods used, and the main outcomes. 2) Make it shorter please.
Introduction:
Although the paper shows a very low number of citations (45 only) Reference #6 is used 7 times which means 15-16%! Which is uncommon.
Line# 76: (George et al., submitted). ? is this a submitted work not yet published? Why it should be here and in that form without listing and order the same as other references? Does the policy of the journal accept such citations? Please review the policy.
Very poor literature review, a decent literature review should include about 20 modern literature, author/s should develop the introduction section significantly.
Materials and Methods:
Do not leave the title without text, please write under the Materials and Methods a short paragraph to explain generally the materials and methods used then zoom in order 2.1. , 2.2….. etc the same for the Results section, please.
The author/s didn’t justify why they use the approach of Objective 2: CFIR or objective2: RANAS. Why do you believe this approach is better than many others? Please write a short paragraph of justification including the characteristics of these approaches.
The methods don’t clarify whether the research approach considers a qualitative or quantitative approach or both. This is highly recommended to clarify also should clarify which is secondary and which once is primary data and how have been used.
What was the timeline for the trial and field work takes to visit the 50 households and 84 individuals? This need to clarify in detail.
Line# 181: “adult over 18 years of age” is this necessary to mention?
2.3. Sampling and Recruitment: doesn’t worth be a separate paragraph, please combine it with the previous one.
2.6. Ethics: same thing here need to combine with the previous one.
Generally, the Material and methods section also needs major changes and restructuring.
Results:
3.4. Subsection line #424 I believe you need to find another title or try to restructure the results section.
Author Response
Reviewer 3
|
Title: Too long need to be changed and make it shorter and keep representing the core value of the work. |
We thank the reviewers for their thoughtful consideration of our manuscript. We have shorted the title as follows: “Implementing an arsenic-mitigation intervention for private well users in American Indian communities: a qualitative evaluation of the Strong Heart Water Study program” |
|
Abstract: The abstract should be rephrased: 1) including, in a nutshell, the hypothesis, objective, the key methods used, and the main outcomes. 2) Make it shorter please. |
We have revised the abstract as below, and reduced word count. The specific requested information is bolded. Currently, the abstract is within the word limit guidelines for IJERPH.
Arsenic is a naturally occurring toxicant in groundwater, which increases cancer and cardiovascular disease risk. American Indian populations are disproportionately exposed to arsenic in drinking water. The Strong Heart Water Study (SHWS), through a community-centered approach for intervention development and implementation, delivered an arsenic mitigation program for private well users in American Indian communities. The SHWS program comprised water arsenic testing, point-of-use arsenic filter installation, and a mobile health program to promote sustained filter use and maintenance (i.e., changing the filter cartridge). Half of enrolled households received additional in-person health communication and videos. Our objectives for this study were to assess implementation successes and barriers and facilitators to implementation, use, and maintenance of the program among implementers and recipients. We conducted 45 semi-structured interviews with implementers and SHWS program households. We analyzed barriers and facilitators using the Consolidated Framework for Implementation Research and the Risks, Attitudes, Norms, Abilities, and Self-regulation model. At the implementer-level, facilitators included building rapport and trust between implementers and participating households. Barriers included remoteness of households, coordinating with plumbers for arsenic filter installation, and difficulty securing a local supplier for replacement filter cartridges. At the recipient-level, facilitators included knowledge of the arsenic health risks, perceived effectiveness of the filter, and visual cues to promote habit formation. Barriers included attitudes towards water taste and temperature and inability to procure or install replacement filter cartridges. This study offers insights on the successes and challenges of implementing an arsenic mitigation program tailored to American Indian households, which can inform future programs in partnership with these and potentially similar affected communities. Our study suggests that building credibility and trust between implementers and participants is important for the success of arsenic-mitigation programs.
In qualitative research, it is common to have an exploratory objective and no hypothesis (see references below), and as such we have not added a hypothesis statement to the abstract.
· Creswell, J. W., & Clark, V. L. P. (2004). Principles of qualitative research: Designing a qualitative study. Office of Qualitative & Mixed Methods Research, University of Nebraska, Lincoln. · Ritchie, J., Lewis, J., Nicholls, C. M., & Ormston, R. (Eds.). (2013). Qualitative research practice: A guide for social science students and researchers. SAGE Publishing. · Hennink, M., Hutter, I., & Bailey, A. (2020). Qualitative research methods. SAGE Publishing.
|
|
Introduction: Although the paper shows a very low number of citations (45 only) Reference #6 is used 7 times which means 15-16%! Which is uncommon. |
We have added the following references to the introduction and methods to strengthen the literature review and justification per reviewer comments:
· Brown, C.H.; Curran, G.; Palinkas, L.A.; Aarons, G.A.; Wells, K.B.; Jones, L.; Collins, L.M.; Duan, N.; Mittman, B.S.; Wallace, A. An Overview of Research and Evaluation Designs for Dissemination and Implementation. Annu Rev Public Health 2017, 38, 1–22. · Carroll, C.R.; Noonan, C.; Garroutte, E.M.; Navas-Acien, A.; Verney, S.P.; Buchwald, D. Low-Level Inorganic Arsenic Exposure and Neuropsychological Functioning in American Indian Elders. Environ Res 2017, 156, 74–79, doi:10.1016/J.ENVRES.2017.03.018. · Gribble, M.O.; Howard, B. v.; Umans, J.G.; Shara, N.M.; Francesconi, K.A.; Goessler, W.; Crainiceanu, C.M.; Silbergeld, E.K.; Guallar, E.; Navas-Acien, A. Arsenic Exposure, Diabetes Prevalence, and Diabetes Control in the Strong Heart Study. Am J Epidemiol 2012, 176, 865–874, doi:10.1093/AJE/KWS153. · Kirk, M.A.; Kelley, C.; Yankey, N.; Birken, S.A.; Abadie, B.; Damschroder, L. A Systematic Review of the Use of the Consolidated Framework for Implementation Research. Implementation Science 2015, 11, 1–13. · May, T.W.; Wiedmeyer, R.H.; Gober, J.; Larson, S. Influence of Mining-Related Activities on Concentrations of Metals in Water and Sediment from Streams of the Black Hills, South Dakota. Arch Environ Contam Toxicol 2001, 40, 1–9, doi:10.1007/S002440010142/METRICS. · Mohammed Abdul, K.S.; Jayasinghe, S.S.; Chandana, E.P.S.; Jayasumana, C.; de Silva, P.M.C.S. Arsenic and Human Health Effects: A Review. Environ Toxicol Pharmacol 2015, 40, 828–846, doi:10.1016/J.ETAP.2015.09.016. · Naujokas, M.F.; Anderson, B.; Ahsan, H.; Vasken Aposhian, H.; Graziano, J.H.; Thompson, C.; Suk, W.A. The Broad Scope of Health Effects from Chronic Arsenic Exposure: Update on a Worldwide Public Health Problem. Environ Health Perspect 2013, 121, 295–302, doi:10.1289/EHP.1205875. · Navas-Acien, A.; Umans, J.G.; Howard, B. v.; Goessler, W.; Francesconi, K.A.; Crainiceanu, C.M.; Silbergeld, E.K.; Guallar, E. Urine Arsenic Concentrations and Species Excretion Patterns in American Indian Communities Over a 10-Year Period: The Strong Heart Study. Environ Health Perspect 2009, 117, 1428–1433, doi:10.1289/EHP.0800509 · Nigra, A.E.; Olmedo, P.; Grau-Perez, M.; O’Leary, R.; O’Leary, M.; Fretts, A.M.; Umans, J.G.; Best, L.G.; Francesconi, K.A.; Goessler, W.; et al. Dietary Determinants of Inorganic Arsenic Exposure in the Strong Heart Family Study. Environ Res 2019, 177, 108616, doi:10.1016/J.ENVRES.2019.108616. · Sclar, G.D.; Bauza, V.; Bisoyi, A.; Clasen, T.F.; Mosler, H.J. Contextual and Psychosocial Factors Influencing Caregiver Safe Disposal of Child Feces and Child Latrine Training in Rural Odisha, India. PLoS One 2022, 17, e0274069, doi:10.1371/JOURNAL.PONE.0274069. · Setty, K.; Cronk, R.; Setty, K.; Cronk, R.; George, S.; Anderson, D.; O’Flaherty, G.; Bartram, J. Adapting Translational Research Methods to Water, Sanitation, and Hygiene. Int J Environ Res Public Health 2019, 16, doi:10.3390/ijerph16204049. · Spaur, M.; Lombard, M.A.; Ayotte, J.D.; Harvey, D.E.; Bostick, B.C.; Chillrud, S.N.; Navas-Acien, A.; Nigra, A.E. Associations between Private Well Water and Community Water Supply Arsenic Concentrations in the Conterminous United States. Science of The Total Environment 2021, 787, 147555, doi:10.1016/J.SCITOTENV.2021.147555. · Bauer, M.S.; Kirchner, J. Implementation Science: What Is It and Why Should I Care? Psychiatry Res 2020, 283, 112376, doi:https://doi.org/10.1016/j.psychres.2019.04.025. · Nilsen, P. Making Sense of Implementation Theories, Models and Frameworks. Implementation Science 2015, 10, 1–13, doi:10.1186/S13012-015-0242-0/TABLES/2.
This manuscript is a process evaluation of a randomized controlled trial. Reference #6 is the paper describing the intervention design. This is why it is cited so many times. |
|
Line# 76: (George et al., submitted). ? is this a submitted work not yet published? Why it should be here and in that form without listing and order the same as other references? Does the policy of the journal accept such citations? Please review the policy. |
This is the main impact evaluation manuscript for the randomized controlled trial, for which this present study reports process and implementation outcomes. This citation provides context to readers about the intervention being evaluated. Citing submitted work is within the journal policy. We have added details for this reference to the bibliography. |
|
Very poor literature review, a decent literature review should include about 20 modern literature, author/s should develop the introduction section significantly. |
We have added the following references to the introduction and methods to strengthen the literature review and justification per reviewer comments:
· Brown, C.H.; Curran, G.; Palinkas, L.A.; Aarons, G.A.; Wells, K.B.; Jones, L.; Collins, L.M.; Duan, N.; Mittman, B.S.; Wallace, A. An Overview of Research and Evaluation Designs for Dissemination and Implementation. Annu Rev Public Health 2017, 38, 1–22. · Carroll, C.R.; Noonan, C.; Garroutte, E.M.; Navas-Acien, A.; Verney, S.P.; Buchwald, D. Low-Level Inorganic Arsenic Exposure and Neuropsychological Functioning in American Indian Elders. Environ Res 2017, 156, 74–79, doi:10.1016/J.ENVRES.2017.03.018. · Gribble, M.O.; Howard, B. v.; Umans, J.G.; Shara, N.M.; Francesconi, K.A.; Goessler, W.; Crainiceanu, C.M.; Silbergeld, E.K.; Guallar, E.; Navas-Acien, A. Arsenic Exposure, Diabetes Prevalence, and Diabetes Control in the Strong Heart Study. Am J Epidemiol 2012, 176, 865–874, doi:10.1093/AJE/KWS153. · Kirk, M.A.; Kelley, C.; Yankey, N.; Birken, S.A.; Abadie, B.; Damschroder, L. A Systematic Review of the Use of the Consolidated Framework for Implementation Research. Implementation Science 2015, 11, 1–13. · May, T.W.; Wiedmeyer, R.H.; Gober, J.; Larson, S. Influence of Mining-Related Activities on Concentrations of Metals in Water and Sediment from Streams of the Black Hills, South Dakota. Arch Environ Contam Toxicol 2001, 40, 1–9, doi:10.1007/S002440010142/METRICS. · Mohammed Abdul, K.S.; Jayasinghe, S.S.; Chandana, E.P.S.; Jayasumana, C.; de Silva, P.M.C.S. Arsenic and Human Health Effects: A Review. Environ Toxicol Pharmacol 2015, 40, 828–846, doi:10.1016/J.ETAP.2015.09.016. · Naujokas, M.F.; Anderson, B.; Ahsan, H.; Vasken Aposhian, H.; Graziano, J.H.; Thompson, C.; Suk, W.A. The Broad Scope of Health Effects from Chronic Arsenic Exposure: Update on a Worldwide Public Health Problem. Environ Health Perspect 2013, 121, 295–302, doi:10.1289/EHP.1205875. · Navas-Acien, A.; Umans, J.G.; Howard, B. v.; Goessler, W.; Francesconi, K.A.; Crainiceanu, C.M.; Silbergeld, E.K.; Guallar, E. Urine Arsenic Concentrations and Species Excretion Patterns in American Indian Communities Over a 10-Year Period: The Strong Heart Study. Environ Health Perspect 2009, 117, 1428–1433, doi:10.1289/EHP.0800509 · Nigra, A.E.; Olmedo, P.; Grau-Perez, M.; O’Leary, R.; O’Leary, M.; Fretts, A.M.; Umans, J.G.; Best, L.G.; Francesconi, K.A.; Goessler, W.; et al. Dietary Determinants of Inorganic Arsenic Exposure in the Strong Heart Family Study. Environ Res 2019, 177, 108616, doi:10.1016/J.ENVRES.2019.108616. · Sclar, G.D.; Bauza, V.; Bisoyi, A.; Clasen, T.F.; Mosler, H.J. Contextual and Psychosocial Factors Influencing Caregiver Safe Disposal of Child Feces and Child Latrine Training in Rural Odisha, India. PLoS One 2022, 17, e0274069, doi:10.1371/JOURNAL.PONE.0274069. · Setty, K.; Cronk, R.; Setty, K.; Cronk, R.; George, S.; Anderson, D.; O’Flaherty, G.; Bartram, J. Adapting Translational Research Methods to Water, Sanitation, and Hygiene. Int J Environ Res Public Health 2019, 16, doi:10.3390/ijerph16204049. · Spaur, M.; Lombard, M.A.; Ayotte, J.D.; Harvey, D.E.; Bostick, B.C.; Chillrud, S.N.; Navas-Acien, A.; Nigra, A.E. Associations between Private Well Water and Community Water Supply Arsenic Concentrations in the Conterminous United States. Science of The Total Environment 2021, 787, 147555, doi:10.1016/J.SCITOTENV.2021.147555. · Bauer, M.S.; Kirchner, J. Implementation Science: What Is It and Why Should I Care? Psychiatry Res 2020, 283, 112376, doi:https://doi.org/10.1016/j.psychres.2019.04.025. · Nilsen, P. Making Sense of Implementation Theories, Models and Frameworks. Implementation Science 2015, 10, 1–13, doi:10.1186/S13012-015-0242-0/TABLES/2.
|
|
Materials and Methods: Do not leave the title without text, please write under the Materials and Methods a short paragraph to explain generally the materials and methods used then zoom in order 2.1. , 2.2….. etc the same for the Results section, please. |
We have added the following text under the main heading “2. Methods and Materials”: “In this study, we applied qualitative methods to data collected through semi-structured interviews with implementers and recipients. We apply qualitative frameworks from implementation and behavioral science [21-23] to evaluate successes, barriers and facilitators to intervention uptake. Specific methods are described below.” We have double checked all headings in the methods and results to ensure that they are properly formatted with headings, and corrected the instance (section 3.4) where a title was missing. |
|
The author/s didn’t justify why they use the approach of Objective 2: CFIR or objective2: RANAS. Why do you believe this approach is better than many others? Please write a short paragraph of justification including the characteristics of these approaches. |
Thank you for your suggestion. We have added the following justification: “Both the CFIR and RANAS model have been applied in previous water, sanitation, and hygiene (WaSH) research to describe the factors that influence implementation [24–26] and behavior change [27–32]. These frameworks are specifically designed to examine the factors determining implementation and household uptake. Application of conceptual frameworks allows for more comprehensive evaluation and provides a structure with which to categorize and examine relationships between factors, which would otherwise be challenging. They also promote reproducibility and comparison of findings across studies of similar interventions in other contexts, as they provide well established definitions of constructs [33,34].”
|
|
The methods don’t clarify whether the research approach considers a qualitative or quantitative approach or both. This is highly recommended to clarify also should clarify which is secondary and which once is primary data and how have been used. |
We have added the following text under the main heading “2. Methods and Materials”: In this study, we applied qualitative methods, using semi-structured interviews with implementers and recipients. We apply conceptual frameworks from implementation and behavioral science [21–23] to evaluate successes, barriers, and facilitators to implementation and intervention uptake. Specific methods are described below.
As this research is purely quantitative, the distinction between primary and secondary outcomes is not applicable, as qualitative research does not make this distinction.
|
|
What was the timeline for the trial and field work takes to visit the 50 households and 84 individuals? This need to clarify in detail. |
We have revised the methods section to now specify the months in which data were collected: “Two authors (KE, RS) conducted the preliminary semi-structured interviews in December 2018 and December 2021. A single author (DA) conducted all subsequent interviews in May 2022. The SHWS RCT was conducted from July 2018 to May 2021”
|
|
Line# 181: “adult over 18 years of age” is this necessary to mention? |
We find this information is necessary to mention, as if we just say “adults” without specifying an age the reader will not know what specific ages were eligible to participate in this study. |
|
2.3. Sampling and Recruitment: doesn’t worth be a separate paragraph, please combine it with the previous one. |
Including sampling and recruitment as an independent section is extremely common in qualitative research. See recent IJERPH publications e.g.,
Tomaino et al. 2023: https://www.mdpi.com/1660-4601/20/2/1037 Wilfling et al. 2023 : https://www.mdpi.com/1660-4601/20/2/1033 Tabangcura et al. 2023: https://www.mdpi.com/1660-4601/20/2/1029
We have retained this as a separate section, as we find it improves clarity and navigability of the manuscript for the reader. |
|
2.6. Ethics: same thing here need to combine with the previous one. |
We have referenced style guidelines and examples of work from IJERPH, and it seems the norm is including this information on IRB only in a separate declarations section at the end of the manuscript. Accordingly, we have deleted this information from the main body of the manuscript and retained it in the declarations section at the end. |
|
Generally, the Material and methods section also needs major changes and restructuring. |
We have restructured these sections as described above. |
|
Results: 3.4. Subsection line #424 I believe you need to find another title or try to restructure the results section. |
We have added the correct title of this section as “Barriers and facilitators among recipients" |

Round 2
Reviewer 3 Report
The author/s did significant improvements and answered/revised all the comments. Thanks to the author/s for their time and efforts and wishing them all the best in the future.